# High Dosage of Patient-Controlled Epidural Analgesia (PCEA) with Low Background Infusion during Labor: A Randomized Controlled Trial

**DOI:** 10.3390/jpm13040600

**Published:** 2023-03-29

**Authors:** Yu Wei, Yilong Wang, Yanhong Zhao, Chaomin Wu, Henry Liu, Zeyong Yang

**Affiliations:** 1Department of Anesthesiology, International Peace Maternity and Child Health Hospital, Shanghai Jiao Tong University School of Medicine, Shanghai 200030, China; 2Shanghai Key Laboratory of Embryo Original Disease, Shanghai 200030, China; 3Shanghai Municipal Key Clinical Specialty, Huashan Rd. 1961, Shanghai 200030, China; 4Department of Pulmonary Medicine, Qingpu Branch, Zhongshan Hospital, Fudan University, Shanghai 200032, China; 5Department of Anesthesiology and Critical Care, Perelman School of Medicine, The University of Pennsylvania, 3401 Spruce Street, Philadelphia, PA 19104, USA

**Keywords:** labor analgesia, epidural, low background dose, patient-controlled epidural analgesia, continuous infusion

## Abstract

Background: Patient-controlled epidural analgesia (PCEA) is well documented; however, it is unclear whether a high dosage of PCEA with a low dosage of background infusion during labor can be a safe and effective application. Methods: Group LH was administered a continuous infusion (CI) of 0.084 mL/kg/h with PCEA of 5 mL every 40 min. Group HL was given a CI of 0.028 mL/kg/h and PCEA of 10 mL every 40 min; Group HH was given a CI of 0.084 mL/kg/h and PCEA of 10 mL every 40 min. The primary outcomes were VAS pain score, the number of supplemental boluses, incidence of pain outbreaks, drug dose for pain outbreaks, PCA times, effective PCA times, anesthetic consumption, duration of analgesia, duration of labor and delivery outcome. Secondary outcomes included adverse reactions such as itching, nausea and vomiting during analgesia and neonatal Apgar scores 1 min and 5 min after birth. Results: A total of 180 patients, 60 in each group were randomly assigned to one of three groups included group LH, group HL or group HH. The VAS scores were obviously decreased in HL group and HH group in comparison with LL group at 2 h after analgesia and the time point of full cervical dilation and delivery of baby. The time for third stage of labor in HH group was increased compared with LH group and HL group. Incidence of pain outbreaks in LH group was obviously increased compared with HL and HH group. The effective PCA times in HL group and HH group were remarkably reduced compared with those in LH group. Conclusions: High dose of PCEA with a low background infusion can reduce effective PCA times, incidence of outbreak pain and the total amount of anesthetics without diminishing analgesia effects. However, high dose of PCEA with a high background infusion can enhance analgesia effect but increase the third stage of labor, instrumental delivery ratio and the total amount of anesthetics.

## 1. Introduction

While epidural labor analgesia is currently recognized as the safest and most effective type of labor analgesia, patient-controlled epidural analgesia (PCEA) combined with continuous infusion of local anesthetics and opioids (PCEA + CEI) is the most common type of drug administration. Manual bolus administration by PIEB (programmed intermittent epidural bolus) or CEI (continuous epidural infusion) is recognized to provide satisfactory analgesic effect and reduce the accumulation of the toxicity of local anesthetics; however, manual injection is time-consuming and laborious, and it is not realistic to popularize it in the clinic. The safety of PIEB in labor analgesia has not been confirmed for a long time. The greatest impediment to the implementation of PIEB analgesia is the lack of readily available epidural pumps designed to deliver timed boluses or time boluses with PCEA. The two-pump system we used is not clinically practical. 

Further studies are warranted to determine whether this technique has benefit in other clinical applications of epidural pain management. Outbreaks of pain are very common during labor analgesia. Programmed intermittent epidural bolus (PIEB) has demonstrated many advantages compared with background epidural infusion concerning breakthrough pain, by adding PIEB to background epidural infusion + PCEA improved labor analgesia by obviously decreasing the needs of rescue analgesia and extending the effect of sufficient analgesia. This combination led to a higher consumption of local anesthetic with no significant clinical complications [1].

Previous studies have showed that PIEB can increase maternal satisfaction score and decrease the consumption of epidural drugs compared with a continuous epidural infusion [2,3,4,5], which may be attributed to the extensive diffusion of epidural anesthetic when delivered as a bolus rather than continuous infusion [6]. The primary goal of labor analgesia is to achieve an ideal and desired level of pain relief and more satisfactory care provided to the patients. Maternal satisfaction is a multidimensional measure, is a common assessment method involving personal expectations, labor pain, perceived emotional control, communication skills and maternal involvement in decision-making [5]. Therefore, we expect to change the analgesic mode of CEI+PCEA parameters; the new mode can obtain a satisfactory analgesic effect, and does not increase the accumulation of local anesthetics toxicity, labor time and instrumental delivery rate, and other complications. In recent years, there have been discrepancies regarding the dosage of background infusion used in analgesic pumps settings [7].

The current ASA Practice Guidelines [8] do not provide the clinician with a clear conclusion as to whether PCEA should be combined with a continuous infusion. A Cochrane review meta-analysis [9] found a lower incidence of assisted vaginal delivery in women randomly assigned to receive combined spinal-epidural analgesia compared with traditional (high-dose) epidural analgesia, but combined spinal-epidural analgesia did not perform better compared with low-dose epidural regimens. It is the responsibility of the anesthesiologist to minimize the pain according to anesthetic use during maternal delivery. However, the clinical effects of administrating a high dosage of patient-controlled epidural analgesia (PCEA) with a low dosage of background infusion during labor remains ambiguous, so we hypothesized that high dose of PCEA (10 mL) with a low background infusion (HL group, 0.028 mL/kg/h) is superior to lower dose of PCEA (LH group, 5 mL) or high dose of PCEA (HH group, 10 mL) with a high background infusion (0.084 mL/kg/h).

## 2. Materials and Methods

### 2.1. Subjects

This study was approved by the Ethics Committee of the International Peace Maternal and Child Health Hospital Shanghai Jiaotong University School of Medicine (The trial was registered at www.chictr.org.cn (Registration number: ChiCTR1800017833, accessed on 16 August 2020). Written consent was obtained from all participating parturients. The 193 parturients were randomly assigned by a computer-generated list to one of three groups, group LH, group HL or group HH, which was included in a total of 180 patients. Sixty were in each group; thirteen women did not receive allocated intervention.

Exclusion criteria: (1). VAS was still more than 3 at 30 min after the first dose; (2). Dural perforation, analgesia termination due to epidural catheter prolapsing into the bloodstream.

All groups received an initial epidural drug before connecting to the PCEA pump. Group (LH) was administered a 5 mL PCEA (interval time 40 min) bolus with a continuous background infusion of 0.084 mL/kg/h via an analgesic pump. Group (HL) was given a 10 mL PCEA (interval time 40 min) bolus with a continuous infusion of 0.028 mL/kg/h. Group (HH) was given a 10 mL PCEA (interval time 40 min) bolus with a continuous infusion of 0.084 mL/kg/h based on the previous study [10]. Parturients that met the requirements below were included in this study: eligible for vaginal delivery and requesting labor analgesia, ages 20 to 45 years old, weighing 50–100 kg, ASA physical status of II, pregnancy ≥ 36 weeks, primipara, single pregnancy, head position, cervical dilation of 2~3 cm. Those with one or more of the following conditions were excluded from this study: contraindications for intraspinal anesthetic, prenatal application of analgesics (such as tramadol or meperidine), drug history of sedative hypnotics, history of neuropsychiatric disorders, high-risk pregnancy (included placental abruption, placenta previa and severe preeclampsia).

### 2.2. Procedure

After the parturients entered the delivery room for labor, the upper extremity intravenous channel was established, infusing a 10 mL·kg^−1^·h^−1^ lactated Ringer’s solution. The anesthetic solution mixture used in this study was prepared beforehand according to the following formula: dissolve 0.75% ropivacaine (AstraZeneca batch number: LBDX) 100 mg and fentanyl injection (Yichang Humanwell Pharmaceutical Co., Ltd., Yichang, China, batch number: 1170606) 0.2 mg in saline to form a 100 mL mixture containing 0.1% ropivacaine and 2 μg/mL fentanyl. The maternal heart rate, blood pressure, fetal heart rate and contraction intensity were monitored. After routine disinfection and local anesthetic, an 18G epidural catheter was inserted into the epidural space at the L2~3 or L3~4 interspace with the parturient lying down in the left lateral position. The epidural catheter was placed 4–5 cm towards the cephalic side, ensuring no blood or cerebrospinal fluid reflux; a test dose of 2% lidocaine 3 mL with 1:200,000 adrenaline was administered. After a 5 min observation showing that no general spinal anesthesia or extensive epidural block occurred, 10 mL of the mixture was injected via the epidural catheter, with 3–5 mL each time. If the VAS score of the parturient exceeded 3 after 10 min, an additional 5 mL bolus would be administered. If the parturient continued to have inadequate analgesia (VAS > 3), she would be excluded from this study and the anesthetic solution would be replaced with 0.15% ropivacaine plus 2 μg/mL fentanyl for better analgesic effects. If in the process of pushing the injection, the maternal patient had dizziness, nausea and other discomforts, the case should be eliminated. Loss of temperature sense was tested by rubbing alcohol on the skin; when the maternal temperature sense loss reached T10, T6 with VAS < 3, the Ogilvy & Mather Analgesic Pump (AM-3200) with the anesthetic solution intact would be connected. The parturients were told to press the demand button for analgesics when the VAS score exceeded 3. Drug administration would be terminated after the third stage of labor. The demand for drugs was considered invalid if the parturient pressed the button within 40 min of the previous dose. If the demand button was pressed twice within 20 min without pain relief, it was recorded as an incidence of pain outbreak. To ease the pain during the subsequent labor process, 0.1% ropivacaine plus can be manually injected. If the VAS was still greater than 3 scores after 10 min, the anesthesiologist would administer an additional 5–10 mL of 0.2% ropivacaine plus 2 μg/mL fentanyl. If during the analgesia process, the fetus heart > 180 bpm, or <100 bpm, or if late fetal heart deceleration, mutation deceleration or (and) baseline lack of variation occurs, suspend epidural injection and observe for more than 30 min, and then determine the cause of fetal heart mutation.

Random numbers were placed in airtight envelopes, which were sealed; this was done by an anesthesiologist who was not involved with data analysis. Before inducing anesthesia, the same anesthesiologist who performed the randomization opened the envelopes in sequence. The clinical investigators, data collectors and patients involved in the study were blinded to the experimental group assignments and the drug-randomization sequence.

### 2.3. Laboratory Measurements

VAS scores and the modified Bromage scale (0 = bilateral sustained straightening of leg, 1 = unable to straighten leg, 2 = just able to flex knees, 3 = foot movement only) of the parturients were assessed as the primary outcomes before analgesia, 10 min-, 30 min-, 1 h-, 2 h-post analgesia, during full cervical dilation and delivery. The number of supplemental boluses, incidence of pain outbreaks, drug dose for pain outbreaks, PCA times, effective PCA times, anesthetic consumption, duration of analgesia (beginning from self-implementation of labor epidural analgesia to the end of the third stage of labor), duration of labor and delivery outcome were recorded as the primary outcomes. Adverse reactions such as itching, nausea and vomiting during analgesia and neonatal Apgar scores 1 min and 5 min after birth were noted as the secondary outcomes.

### 2.4. Ethics

The study was approved by the Ethics Committee of the International Peace Maternal and Child Health Hospital Shanghai Jiao Tong University School of Medicine, and all participants provided written informed consent for themselves and their infants.

### 2.5. Statistical Analysis

Data were processed using SPSS 22.0 (IBM, Chicago, IL, USA) statistical software. The significance level was set at α = 0.05, and the power was 1–β = 0.8. Using PASS version 22.0 software for the analysis with 60 participants in each group, a possible 10% missing rate was considered. Thus, a total of 200 patients were included in this study. Numerical data that conformed to the normal distribution were expressed as mean ± standard deviation. Analysis of variance (ANOVA) with post hoc test was used for comparison among groups, and the count data were analyzed using the χ^2^ test or the Fisher exact probability method. *p* < 0.05 was considered statistically significant.

## 3. Results

A total of 200 women met the inclusion criteria for this study. Of these, the participants were randomized and 180 women received statistical analysis (Figure 1).

Seven women did not receive allocated intervention because of those patients suffering from some diseases including contraindictions for intraspinal anesthetic (*n* = 2); history of neuropsychiatric disorders(*n* = 2); high-risk pregnancy(*n* = 3).

The differences of mean age, height, weight and weeks of pregnancy among group LH, HL and HH were statistically insignificant (Table 1).

The VAS values of post analgesia as the primary outcomes in three groups were obviously decreased compared with the values before analgesia (*p* < 0.001). The VAS values were obviously decreased in HL group in comparison with LH group at 2 h after analgesia (*p* < 0.01). Meanwhile, compared with LH group, the VAS values in HL and HH group were significantly reduced at the time point of full cervical dilation and delivery of baby (*p* < 0.001). The VAS value in HH group was lowest at the time of delivery of baby among three groups (Table 2).

The time for analgesic duration, first stage of labor and second stage of labor in three groups were of no significance as the primary outcomes (Table 3). However, the time for second and third stage of labor in HL group was reduced in comparison with the time in HH group (## *p* < 0.01). The times for PCA boluses and effective times for PCA boluses in LH group were significantly higher than those in HL and HH groups (*** *p* < 0.001, $$$ *p* < 0.001, respectively). Cumulative amount of epidural infusion doses in HL group was lowest among the three groups (*p* < 0.001). Incidence of pain outbreaks in HL and HH group was obviously decreased compared with LH group. The PCA times and the effective PCA times in HL and HH group were remarkably reduced compared with those in LH group.

There were no significances in three groups for Cesarean ration; meanwhile, instrumental and vaginal delivery ratio as the secondary outcomes were increased in HH group compared with HL group (Table 4).

## 4. Discussion

The timing of labor analgesia has been debated by researchers around the world. Studies have shown that as long as the parturient was proven eligible for vaginal delivery by the obstetrics, when showing signs of regular contraction and normal cervix activities, labor epidural analgesia could be administered if required. However, prolonged epidural labor analgesia could cause anesthetics to reside in the epidural space, which may lead to local anesthetic and opioid accumulation. An excessive amount of local anesthetics can cause neural motor blockade, which in turn could reduce maternal exercise capacity, weakening the pelvic muscles and the force of “pushing down” during the second stage of labor, which may lead to dystocia or aided childbirth [4]. Our study showed that, compared to the traditional mode of drug administration, HL group with a low background infusion (0.028 mL/kg/h) combined with a high-dose PCEA (10 mL, once every 40 min interval) could reduce anesthetic consumption and the demand for supplement boluses without lessening the analgesic effect, hence lowering the risk of excessive local anesthetics and opioid accumulation. The VAS value in HH group with high background infusion (0.084 mL/kg/h) combined with a high-dose PCEA (10 mL, 40 min interval) was lowest at the time of delivery of baby among three groups. However, instrumental delivery ratio was increased in HH group.

The mechanism of low-dose background infusion combined with high-dose PCEA was similar to that of epidural intermittent injection. The high pressure from the PCEA injection allowed the drug to be rapidly injected into the epidural space through the anterior and lateral orifice of the epidural catheter. The anesthetic would then be evenly distributed and more suited for the individual, thus improving the analgesia effects, reducing the consumption of anesthetics, shortening the second stage of labor compared with LH group, and avoiding anxiety from contractions that could have a negative impact on the newborn. This technique would not influence the analgesic effect during labor, nor would it affect the progress of labor. The high pressure from the infusion widens the diffusion range, inducing a higher level of anesthesia to alleviate the pain, resulting in a lower VAS score from the parturient. In recent years, there have been several research works regarding drug administration for labor analgesia, but the results were inconsistent. Some scholars indicated that no conclusion can be drawn regarding the risks or benefits of adding a continuous background infusion to PCEA compared with PCEA-only epidural labor analgesia [11].

There are still controversies in terms of the optimal anesthetic concentration, delivery volume, administration interval and administration mode (on-demand or regular). Although the results in such studies are not completely consistent, one element is basically the same; the effect of epidural intermittent analgesic administration is generally better than high-dose continuous infusions [12]. In a study on intermittent epidural labor analgesia [13], parturients were administered either 2.5 mL analgesia every 15 min, 5 mL every 30 min, or 10 mL (once every hour). The results showed that while the consumption of local anesthetics declined as the time lapsed, maternal comfort and satisfaction remained unchanged, which was consistent with the results in our study.

As for whether a background infusion dose was needed, studies have shown that PCEA+CI provides better analgesia effects with significant reduction in the incidence of pain outbreaks during labor compared to only PCEA. In a previous study, compared with background PCEA (5 mL bolus, 10–12 min locking interval, and 5–10 mL/h infusion), PCEA required only (5 mL bolus, 15 min locking interval) resulted in reduced local anesthetic consumption but increased incidence of breakthrough pain, higher pain score, shorter duration of effective analgesia and lower satisfaction level for parturients [10]. Cervical examinations were administrated by the obstetric doctor every four hours and with the discretion of the obstetrician. They did not have regular cervical examinations more frequently, because they were concerned about maternal comfort. The lack of regular cervical examinations may have affected the documentation of the first and second stage deliveries. The reduced need for clinician bolus supplementation also reduced the workload of anesthesiologists in delivery suites. However, increasing the background infusion rate from 5 to 10 mL/h did not show any clinically significant advantage, and with longer second stage of labor. In our study, incidence of pain outbreaks in HL group and HH group were obviously decreased compared with LH group; the PCA times and the effective PCA times in HL group and HH group were remarkably reduced compared with those in LH group.

In recent years, researchers have become focused on a new type of labor analgesia technique called programmable intermittent epidural bolus (PIEB). McKenzie et al. [14] believed that in comparison with continuous epidural infusion (CEI), recent evidence showed that programmed intermittent epidural bolus (PIEB) improved maternal outcomes, which encouraged us to change our labor epidural analgesia protocols [15]. Programmable intermittent epidural analgesia and continuous epidural analgesia are the two important technical methods of labor analgesia, and the different administration methods can induce different effects on the outcome of the mother and the baby. The speed at which the infusion bolus is delivered and the pressure generated in the epidural space also reflected the difference. Empirically, intermittent boluses injected at higher pressure should add more widespread and uniform epidural solution dispersion [16,17]. Experimentally, the use of intermittent boluses had been found to result in a greater spread of infusion in comparison with a continuous infusion, despite a similar rate of infusion [5]. In vitro studies [18], it was confirmed that when a constant rate of 10.5 mL/h is used for continuous administration, most of the drug solution flows out through the proximal hole of the spinal epidural catheter, and when a single injection is used, both the proximal and distal holes of the lumen catheter of the epidural have liquid outflow, suggesting that when the same dose is taken, a single injection will have a wider range of drug block. It may also be that programmed epidural analgesia is better than continuous epidural analgesia [19,20]. PIEB could reduce the frequency of PCEA boluses and the incidence of pain outbreak during labor, hence improving the quality of analgesia. However, PIEB showed an increase in the level of thoracic sensory block, but a decrease in the level of lumbosacral block. Other specialists had administrated some relevant trials to optimize our PIEB regimen for labor analgesia via changing bolus intervals, bolus volumes and drug concentration. None of these changes decreased the incidence of high sensory block levels without compromising the analgesic effect. PIEB was related to the reduction of Cesarean section rate, but there was no significant difference in maternal satisfaction, motor block or instrumental delivery rate between the two groups. However, the hourly dose of local anesthetic is different between the two groups, the administration of breakthrough pain is not normal and only continuous epidural infusion recipients can obtain patient-controlled epidural analgesia. Patients with a planned intermittent epidural bolus asked the caregivers to perform a manual bolus for breakthrough pain. There is no information about the need for additional rescue analgesia [21,22,23,24]. Another team could not find significant clinical differences from the delivery speeds [25]. Faster delivery rates and higher administration pressures may lead to more extensive spread of local anesthetic into the epidural space. The distribution of anesthetic in the epidural space is affected by many factors [25,26,27,28,29]. A previous study also showed that the high residual epidural pressure rather than the peak pressure decided the upper level of epidural analgesia [30].

Therefore, the safety of the application of PIEB during the whole process of labor, and its corresponding drug concentration and analgesic pump settings, remains to be further studied. We have acknowledged the limitations in this study. For instance, the sample size of this study is relatively small, and we failed to observe the whole process of labor analgesia. The settings of the analgesic pump in this study (including background dose, PCEA dose, lockout interval, etc.) were set according to our own experience. Although the results show that anesthetic consumption had significantly decreased without weakening the analgesic effects, it still may not be the optimal setting. For Chinese parturients, there is a risk that the level of anesthesia from 10 mL of PCEA may be too high for some individuals even if no obvious cardiovascular events or complications occurred such as hypotension and so on, Studies with larger sample size are needed to further investigate the optimal settings for a full-course labor analgesia.

## 5. Conclusions

A low background infusion (0.028 mL/kg/h) combined with a high-dose PCEA (10 mL, 40 min interval) is a safe and effective clinical application for labor analgesia, which reveals an obvious advantage over high background infusion (0.084 mL/kg/h) combined with a high-dose PCEA (10 mL, 40 min interval) by decreasing instrumental delivery ratio, PCA times, effective PCA times, incidence of outbreak pain and the total amount of anesthetics without reducing analgesia effects. However, regardless of the analgesic method, rigorous monitoring, active and effective interventions by clinical doctors are still an indispensable part of a safer and ideal labor analgesia. Clinical strategies will be patient-oriented and constantly improve the clinical safety of maternal and infant patients and the satisfaction of parturients and their families.

## Figures and Tables

**Figure 1 jpm-13-00600-f001:**
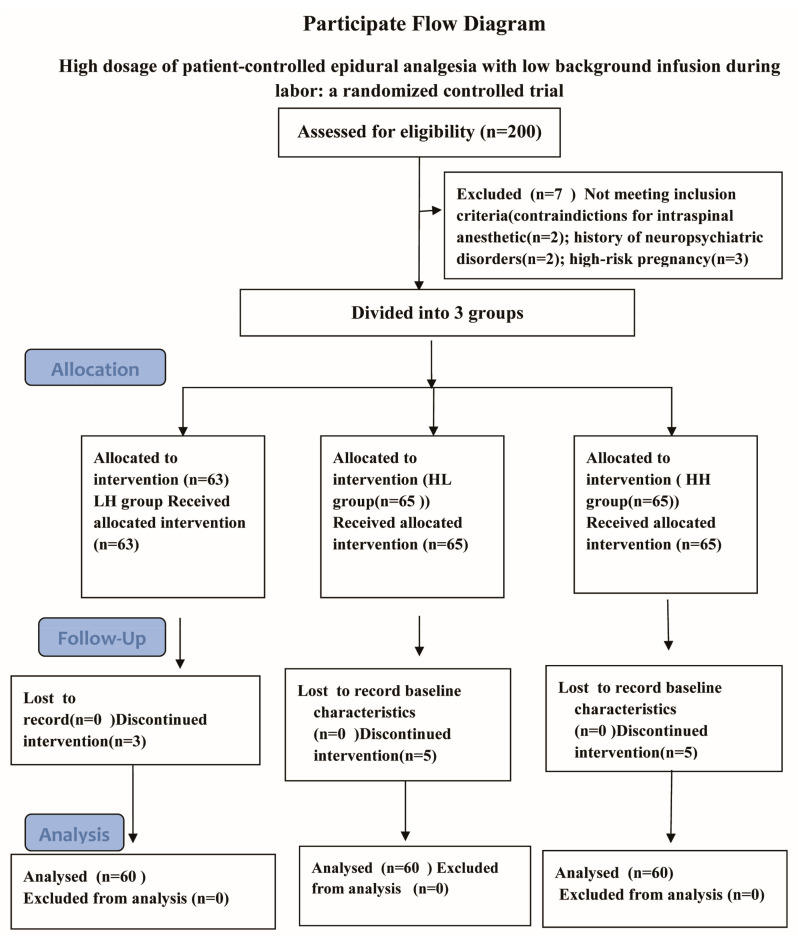
A total of 200 women met inclusion criteria for this study. These were randomized and 180 women received statistical analysis (Figure 1, flow diagram of study).

**Table 1 jpm-13-00600-t001:** Comparison of general characteristics of three groups.

Group	Number of Cases	Age (Year)	Height (cm)	Weight (cm)	Weeks of Pregnancy
LH	60	31.4 ± 4.9	162.5 ± 5.1	71.8 ± 10.2	38.6 ± 1.2
HL	60	31.8 ± 4.5	162.8 ± 5.0	71.9 ± 9.6	38.6 ± 1.1
HH	60	32.3 ± 4.2	163.1 ± 4.2	71.3 ± 8.1	38.6 ± 1.2

Numerical data were expressed as mean ± standard deviation. There were no significant changes (*p* > 0.05).

**Table 2 jpm-13-00600-t002:** Comparison of VAS scores among the three groups.

Group	Number of Cases	Before Analgesia	Post Analgesia				Delivery of Baby
			30 min	1 h	2 h	Full Cervical Dilation	
LH	60	8.39 ± 1.12 &&&	2.03 ± 1.06	2.20 ± 1.16	2.80 ± 1.48 **&&	3.27 ± 1.64 ***$$$	4.17 ± 2.00 ***$$$
HL	60	8.30 ± 1.89 &&&	2.15 ± 1.02	1.95 ± 0.95	2.08 ± 1.12	2.23 ± 1.09	2.88 ± 1.40 ###
HH	60	8.53 ± 1.03 &&&	2.05 ± 1.03	1.85 ± 1.01	2.10 ± 1.15	2.08 ± 1.09	2.13 ± 1.13

Numerical data were expressed as mean ± standard deviation. In comparison with other groups, &&& *p* < 0.001, && *p* < 0.01 (Before analgesia vs. Post analgesia), ** *p* < 0.01 (group LH vs. group HL), *** *p* < 0.001 (group LH vs. group HL), ### *p* < 0.001(group HL vs. group HH), $$$ *p* < 0.001 (group LH vs. group HH) among three groups.

**Table 3 jpm-13-00600-t003:** Comparison of duration of analgesia, stage of labor, PCEA boluses, and drug consumption among the three groups.

Monitoring Index	Group LH	Group HL	Group HH
Analgesic duration (min)	405.32 ± 120.75	383.83 ± 150.60	397.12 ± 82.56
First stage of labor (min)	523.68 ± 248.18	516.07 ± 213.30	560.67 ± 259.53
Second stage of labor (min)	49.55 ± 27.39	45.27 ± 21.42 ##	59.12 ± 30.92
Third stage of labor (min)	7.22 ± 3.96	7.03 ± 3.83 ##	9.82 ± 6.10 $$
Times for PCA boluses	3.32 ± 1.26 ***$$$	1.65 ± 0.84	1.75 ± 0.68
Epidural volume dose	52.56 ± 13.06 ***$$$	37.65 ± 7.56 ###	63.00 ± 8.07
Effective times for PCA boluses	2.50 ± 1.10 ***$$$	1.40 ± 0.69	1.52 ± 0.50
Occurrence rate for outbreak pain	10/60	4/60	3/60 $
Volume dose for outbreak pain	11.29 ± 3.04	8.00 ± 0.00	8.67 ± 1.16

Numerical data were expressed as mean ± standard deviation. The incidence rate of group LH and HL in comparison with Group HH for duration of analgesia, stage of labor, PCEA boluses, and drug consumption among the three groups (*** *p* < 0.001 group LH vs. Group HL; ## *p* < 0.01, ### *p* < 0.001 group HL vs. Group HH; $$$ *p* < 0.001, $$ *p* < 0.01, $ *p* < 0.05 group LH vs. Group HH).

**Table 4 jpm-13-00600-t004:** Comparison of mode of delivery among the three groups [*n* (%)].

Mode of Delivery	Group LH	Group HL	Group HH
Vaginal	43(72) *	44(73) #	31(52)
Instrumental	7(12) *	8(13) #	19(32)
Cesarean	10(16)	8(14)	10(16)

Results were expressed as frequency(proportions). In comparison with Group LH and HL, the incidence rate of group HH was significantly decreased (* or # *p* < 0.05).

## Data Availability

All data generated or analysed during this study are included in this article. Further enquiries can be directed to the corresponding author.

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
