# Peer review of "High Dosage of Patient-Controlled Epidural Analgesia (PCEA) with Low Background Infusion during Labor: A Randomized Controlled Trial"

_jpm, 2023, doi:10.3390/jpm13040600_

Round 1

Reviewer 1 Report

Greetings

I read your manuscript with interest and find the hypothesis very relevant. The study is well conducted and presented; the novelty of the hypothesis and randomization are strengths. However, I feel a few aspects still need gross attention.

Major: Please mention the sample size calculation, allocation concealment and blinding.

Major: Please provide data on hemodynamics, especially in the HH group it is very relevant.

Minor- Pregnant women in labour are not ASA-PS 1; ASA-PS starts from II onwards.

Minor- There is a discrepancy in mentioning the groups. In some places, it is mentioned as group A, B, and C, and in others- based on a high and low dosage (LL, HL, HH). Please correct it.

Minor- In the abstract, the number of participants can be mentioned. 

Best of luck

Author Response

Reviewer1 point to point comment

Greetings

I read your manuscript with interest and find the hypothesis very relevant. The study is well conducted and presented; the novelty of the hypothesis and randomization are strengths. However, I feel a few aspects still need gross attention.

Major: Please mention the sample size calculation, allocation concealment and blinding.

The significance level was set at α = 0.05, and the power was 1–β = 0.8. Using PASS version 22.0 software for the analysis with 60 participants in each group, a possible 10% missing rate was considered.

Random numbers were placed in airtight envelopes, which were sealed; this was done by an anaesthesiologist who was not involved with data analysis. Before inducing anaesthesia, the same anaesthesiologist who performed the randomisation opened the envelopes in sequence. The clinical investigators, data collectors and patients involved in the study were blinded to the experimental group assignments and the drug-randomisation sequence.

Major: Please provide data on hemodynamics, especially in the HH group it is very relevant.

Dear professor, because our study focused on the analgesic effect and relevant complications, the regulation and monitoring of blood pressure by the nursing team was very effective and safe, meanwhile, in this study, our team did not find significant cardiovascular complications. So we didn’t record these data for hemodynamics, which was also the limitations for our experiment.

Minor- Pregnant women in labour are not ASA-PS 1; ASA-PS starts from II onwards.

Dear professor, we have revised them.

Minor- There is a discrepancy in mentioning the groups. In some places, it is mentioned as group A, B, and C, and in others- based on a high and low dosage (LL, HL, HH). Please correct it.

OK! Dear professor, we have corrected them.

Minor- In the abstract, the number of participants can be mentioned. 

Dear professor, we have revised the abstract. RESULTS A total of 180 patients, 60 in each group were randomly assigned to one of three groups included group LL, group HL or group HH.

Best of luck

Reviewer 2 Report

Great work.

Dolantin is generic name so use meperidine in text instead.

Why do you use only lidocaine without epinephrine - how do you now if its in artery/vein ??

It would be wise to implement in your work possible side effects of high volume epidural PCA vs low volume because practitioners always fear of possible side effects and tend to use smaller volumes especially in primipara.

Author Response

Reviewer2 point to point comment

Reviewer2

Great work.

Dolantin is generic name so use meperidine in text instead.

Dear professor, we have revised it.

Why do you use only lidocaine without epinephrine - how do you now if its in artery/vein ??

 We revised them. Lidocaine with 1:200,000 adrenaline.

It would be wise to implement in your work possible side effects of high volume epidural PCA vs low volume because practitioners always fear of possible side effects and tend to use smaller volumes especially in primipara.

Thank you, dear professor, empirically, we monitored the hemodynamics data with real-time.  There are no obvious cardiovascular events or complications occurred just like hypotension and vomiting and so on.